# Electrochemical topological transformation of polysiloxanes

Minami Oka[1] & Satoshi Honda [ID] [1✉]

Coupling reactions between polymers are an important class of chemical modifications for changing, enhancing, and tuning the properties of polymeric materials. In particular, transformation of polymer topologies based on efficient, facile and less wasted coupling reactions remains a significant challenge. Here, we report coupling reactions based on electrochemical oxidation of 2,4,5-triphenylimidazole into a 2,4,5-triphenylimidazolyl radical and its spontaneous dimerization into hexaarylbiimidazole. Based on this chemistry, electrochemical topological transformation (ETT) and electrochemical chain extension have been realized with siloxane-based oligomers and polymers. Moreover, this approach enables one step ETT of star-shaped poly(dimethyl siloxane)s (PDMSs) into network PDMSs, running in an ionic liquid solvent and requiring no purification steps.

---

[1] Department of Basic Science, Graduate School of Arts and Sciences, The University of Tokyo, 3-8-1 Komaba, Meguro-ku, Tokyo 153-8902, Japan.
✉email: c-honda@mail.ecc.u-tokyo.ac.jp

Progress in organic synthesis and polymerization methodologies continues to drive the development of various polymers with nonlinear topologies. Distinctly different properties between linear and nonlinear polymers have created unprecedented opportunities for the production of a broad spectrum of functional materials[1]. In particular, various topological transformation systems based on polymer–polymer coupling reactions have widely been studied[2,3], yet the development of an efficient and greener approach remains challenging.

In this context, polymer–polymer coupling and transformation of topologies of polymers based on the cleavage and reformation of the covalent bond between the two imidazole rings in hexaarylbiimidazole (HABI) upon photostimulation have been reported[4–7]. Conversion of 2,4,5-triphenylimidazole (lophine) into 2,4,5-triphenylimidazolyl radical (TPIR) in the presence of a mild oxidizing agent such as pottasium ferricyanide in the biphasic organic–aqueous system causes spontaneous C–N or N–N dimerization between the pair of TPIRs into HABIs[8], resulting in coupling between polymers (Fig. 1a). The polymers linked with HABIs in the chains are practically important because they display stress-induced color changes[6] and photoreversible dramatic changes in viscoelasticity[4]. Moreover, the remarkable

photoresponse of HABIs and excellent reversibility of the photoreaction find their applications in photochromic materials as studied in detail by Abe and coworkers[9,10]. On another front, HABIs have been utilized as photoradical generators (PRGs) in an industrialized photopolymerization process developed by DuPont[11]. In this process, TPIR generated upon photoirradiation to HABI first reacts with a chain transfer agent such as thiol and then a produced secondary radical species initiates polymerization of vinyl monomers. This concept has recently been extended to thiol–ene reaction system toward dental restorative materials[12]. Further expansion of the range of possible applications is thus intriguing, yet the conversion of lophine into TPIR with the above system requires a wasteful amount of solvent, oxidant, and base. Hence, the development of an alternative approach is a critical issue to be tackled.

To this end, electrochemistry is promising because it is atom-economic and less waste[13]. Historically, electrochemical oxidation of lophine was first examined in 1970[14], yet the concept has not been extended to coupling reaction systems between polymers to date. On the other hand, as represented by the electrochemical N–N dimerization with carbazoles[15], recent rational and systematic exploration for electrochemical reactions has sophisticated the concept and technique for electrochemistry[16,17] and has overcome even the failure of oxidation by chemical reagents. Carbazoles have also been employed as monomers for electrochemial oxidation polymerization[18]. While N–N dimers have been regarded as nonpolymerized byproducts for this purpose, N-alkyl-substituted carbazoles selectively afford linear polycarbazoles[19]. Recently, the concept has been further extended to topologically controllable growth of linear or crosslinked polymers in step growth electrochemial oxidation polymerization of N-alkyl-substituted carbazoles[20]. On the other hand, little progress has been made in the study of the effective utilization of N atoms for relevant dimerization reactions for polymer synthesis. Inspired by the structural similarities between carbazoles and lophines, we initiated work aimed at the electrochemical oxidation of lophine-appended polymers and following C–N and N–N dimerization of the produced TPIRs into HABIs, with the intent of realizing a polymer–polymer coupling and transformation of topologies of polymers (Fig. 1a).

Herein, we report the realization of electrochemical topological transformation (ETT) and electrochemical chain extension (ECE) of oligo(dimethyl siloxane) (ODMS) and poly(dimethyl siloxane) (PDMS) (Fig. 1b). Moreover, we have extended the chemistry toward one step network formation that does not require purification procedures through ETT of star-shaped polymers into their networks (Fig. 1b) by exploiting silicone soluble ionic liquid doubled as solvent and support electrolyte.

## Results and discussion

**Model reaction with small molecular lophine**. To investigate electrochemical coupling, we initiated a model reaction using a small molecule, i.e., bromo-appended lophine (**1**) (Fig. 2a). Both **1** and the product after chemical oxidation (**2**) are known compounds[21] and thus allow straightforward characterization. Screening revealed successful oxidation in the presence of (*n*Bu)$_4$NClO$_4$ as a support electrolyte in dichloromethane (DCM) or in a mixture of DCM and methanol (MeOH). A cyclic voltammogram (CV) of **1** in DCM/MeOH (10/3, v/v) indicated that a potential below +1.3 V is better for the reaction, as an increase in potential resulted in decomposition in the experimental potential range (+0–2.0 V) similar to that reported in the literature (Fig. 2b)[15]. Upon electrochemical oxidation, the color of the solution immediately turned purple (Supplementary Fig. 1), and the electron spin resonance (ESR) spectrum of the product

**(a)**

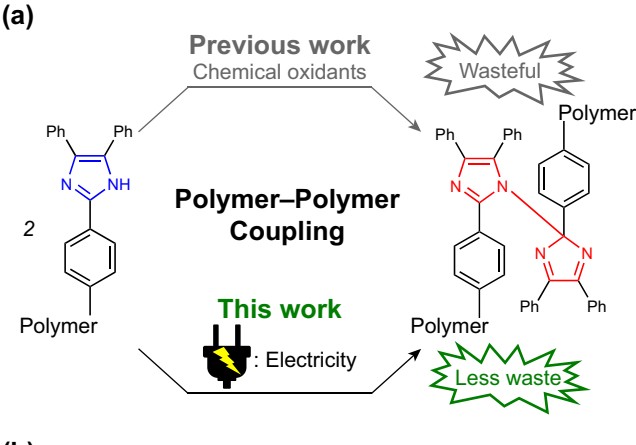

**(b)**

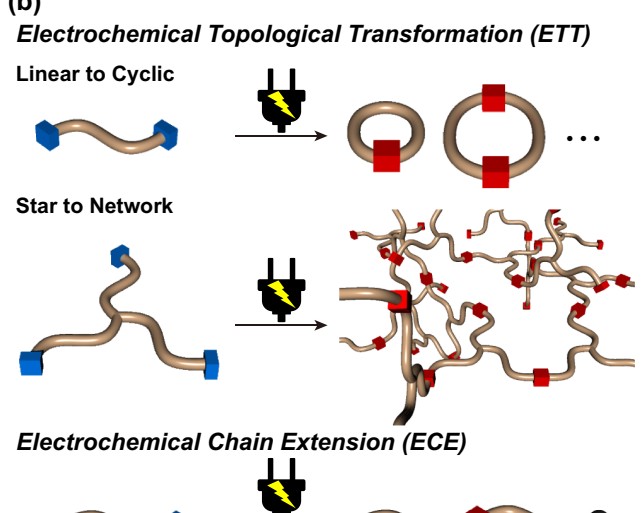

**Fig. 1 Oxidation of lophines and its application for polymer synthesis.**
**a** Previous polymer–polymer coupling reaction based on oxidation of lophines utilizing chemical oxidants and electrochemical polymer–polymer coupling in this work. **b** Schematic illustration of linear to cyclic and star to network electrochemical topological transformations (ETTs) and chain extension (ECE) reactions reported in this work.

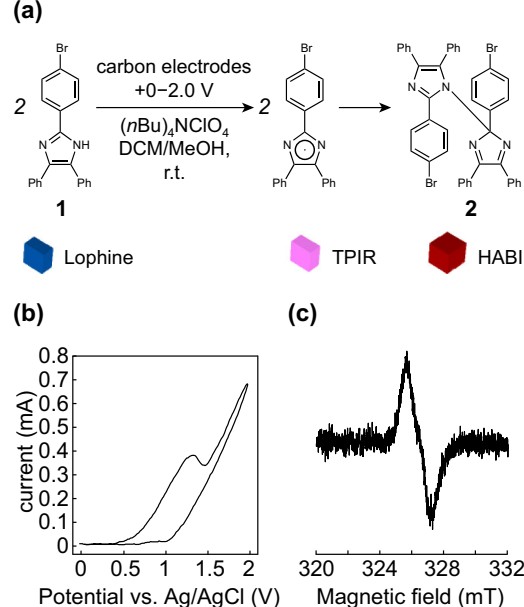

**Fig. 2 Electrochemical oxidation of 1. a** Synthesis of **2** upon applying potential to DCM/MeOH solution of **1** containing ($n$Bu)$_4$NClO$_4$ with carbon electrodes. **b** Cyclic voltammogram of **1** using ($n$Bu)$_4$NClO$_4$ as a support electrolyte. **c** ESR spectrum of **2** upon photoirradiation.

after electrochemical oxidation showed a TPIR-derived signal at approximately 326 mT upon photoirradiation at a wavelength of 405 nm (Fig. 2c). These results strongly support the electrochemical oxidation of lophine into TPIR and subsequent coupling between a pair of TPIRs into HABI, as outlined in Fig. 2a. The production of **2** was confirmed by atmospheric pressure chemical ionization (APCI) mass analysis by the appearance of the peak at $m/z = 747.5$, which is double the peak that appeared with **1** at $m/z = 375.3$ (Supplementary Fig. 2), demonstrating the intended dimerization. Comparison of $^1$H NMR spectra between **1** (Supplementary Fig. 3a) and **2** (Supplementary Fig. 3b) confirmed complete consumption of **1** by the disappearance of the signal at 12.8 ppm derived from –NH proton (Supplementary Fig. 3a). The increased complexity of the signal in the aromatic region suggests the presence of multiple isomers derived from HABI[22,23], which is consistent with the effective conversion of **1** into **2**.

**Electrochemical linear to cyclic topological transformation of monodisperse siloxane oligomers.** Having achieved electrochemical oxidation with the model reaction, we next examined the electrochemical cyclization of monodisperse linear oligo(dimethyl siloxane) (ODMS) with lophine end groups (**M$_L$**). **M$_L$** was synthesized according to a recently reported procedure (Supplementary Scheme 1)[23] and subjected to electrochemical oxidation (Fig. 3a). Initially, we faced a severe restriction of the condition because of their insolubility in polar solvents typically used for the electrochemical oxidation reactions. For instance, the addition of a slight amount of MeOH to a DCM solution of **M$_L$** caused phase separation, which is undesirable due to the inaccessibility of the reacting sites to the electrode interface. Among various solvents for electrochemical oxidation, DCM effectively worked for our PDMSs. Although a considerably low effective concentration of lophine end groups in whole of the material prevented CV analysis, the color of the reaction solution changed slightly to blue upon applying potential to **M$_L$** (Supplementary Fig. 4) owing to the production of ODMS with TPIR end groups (**M$_L$***). The formation of **M$_L$*** was also confirmed based on UV-vis spectrometry by the appearance of TPIR-derived absorption[5,24] upon

photoirradiation to its dimerization product **M$_C$** (Fig. 3b). The production of **M$_C$** was also confirmed by comparing $^1$H NMR spectra between **M$_L$** (Supplementary Fig. 5a) and **M$_C$** (Supplementary Fig. 5b). Similar to the model reaction, the signal derived from –NH proton at 9.8 ppm visible before the reaction completely disappeared from the $^1$H NMR spectrum of **M$_C$** (Supplementary Fig. 5b). In principle, the present electrochemical oxidation could proceed inter- and intramolecularly, and thus, the product composition achieved by the present electrochemical oxidation is intriguing. Comparison of size-exclusion chromatography (SEC) traces between **M$_L$** and **M$_C$** revealed dominant unimolecular cyclization. Thus, the peak molecular weights ($M_p$s) of **M$_L$** and **M$_C$** calculated from their traces were 1700 and 1300, respectively (Fig. 3c). The decrease in apparent molecular weight (**M$_C$**/**M$_L$** = 0.76), i.e., hydrodynamic volume in the solution state analyzed by SEC is strong indicative of the formation of a three-dimensionally more compact cyclic topology[25,26]. The appearance of the shoulder peak with an $M_p$ of 2800 suggests the formation of a dimer. SEC also showed dominant formation of a unimolecularly cyclized product. The electrospray ionization (ESI) high-resolution mass (HRMS) spectrum of **M$_C$** showed a peak at $m/z = 1147.4458$, which exactly matches the calculated molar mass of its $[M + H]^+$ion ($m/z = 1147.4384$) (Fig. 3d). Moreover, the observed isotopic pattern of $[M + H]^+$exactly corresponded with the simulated pattern (Fig. 3d). In our previous work, chemical oxidation of **M$_L$** in a biphasic organic–aqueous system resulted in preferential unimolecular cyclization despite both intermolecular chain extension and intramolecular cyclization being possible for end-to-end reactions between telechelic linear polymers[23]. The preferential unimolecular cyclization is explained by the low concentration of reactive TPIR end groups in the organic phase because the conversion of lophines into TPIRs, in principle, occurs only at the interface of the two phases. Similarly, electrochemical reactions occur at the interfaces of electrodes, and thus, the generation of reactive TPIRs is limited only at the interface. This likely allowed pseudodilution conditions for reactive TPIR end groups, which is favorable for intramolecular cyclization.

**Electrochemical chain extension of polysiloxanes.** The applicability of the present methodology to reactions with much longer molecular chains, i.e., polymer–polymer coupling is evident from electrochemical oxidation of poly(dimethyl siloxane) (PDMS). Linear PDMS with a lophine end group (**P$_L$**) was synthesized in two steps (Supplementary Scheme 2) and subjected to electrochemical oxidation (Fig. 4a). In contrast to **M$_L$** (Supplementary Fig. 4), an initial test conducted inside a vial showed pink coloration of the reaction solution upon applying potential (Fig. 4a, inset photograph). We thus established a homebuilt in situ UV–vis measurement cell for electrochemical oxidation, which was composed of a quartz cell and carbon electrodes equipped with a screw cap with holes for fixing the electrodes as detailed in Supplementary Fig. 6. Time-dependent UV-vis measurements conducted based on this system clearly showed a sudden increase in TPIR-derived adsorption at a wavelength of 568 nm ($A_{568}$) upon applying a potential of 1.25 V at $t = 10$ s (Fig. 4b, inset). Afterward, $A_{568}$ decreased owing to coupling between TPIRs, and the decrease was almost complete within 60 min (Fig. 4b). This change in adsorption is practically important because the completion of the reaction is visually apparent. As the difference in the coloration between **M$_L$** and **P$_L$** is possibly derived from the substitution group of lophine, modifying the chemical structure of lophine might allow tunability of the reaction and its traceability. Similar to the aforementioned $^1$H NMR analyses, it was corroborated that the lophine end group of **P$_L$** was completely

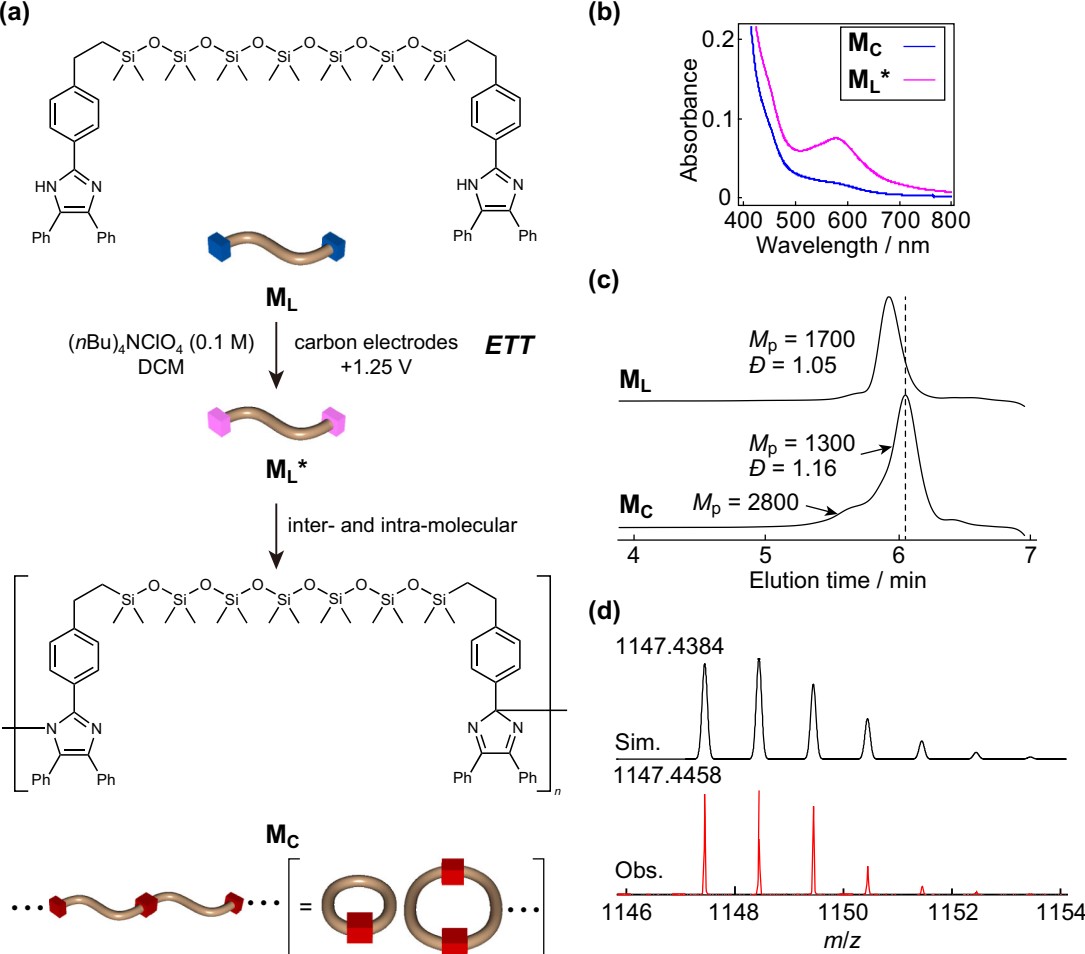

**Fig. 3 Electrochemical oxidation of $M_L$. a** Synthesis of $M_C$ upon applying a potential of 1.25 V to DCM solution of $M_L$ containing ($n$Bu)$_4$NClO$_4$ with carbon electrodes. **b** UV–vis spectra of $M_C$ and $M_L$\*. **c** SEC traces of $M_L$ and $M_C$. **d** Simulated and observed isotopic patterns of $M_C$ analyzed by positive mode ESI-HRMS spectrum.

converted to HABIs from the comparison of $^1$H NMR spectra between $P_L$ (Supplementary Fig. 7a) and $P_D$ (Supplementary Fig. 7b). The SEC trace of the product showed a peak molecular weight ($M_p$) twice as large as that of $P_L$, demonstrating efficient coupling between $P_L$s (Fig. 4c). It should be noted that the purification process—evaporation to remove the solvent and subsequent washing with MeOH to remove the support electrolyte—is quite simple, and both the solvent and support electrolyte can be reusable. This aspect is distinguished from the chemical oxidation process requiring excess amounts of solvent, base, and oxidants.

From the early report of electrochemical oxidation of lophines, a lophine anion was first generated either by treatment using a strong base or by reacting with the radical anion of benzonitrile generated in situ upon electrolysis and then electrochemical oxidation took place to produce its anion radical[14]. Subsequently, these anion radicals underwent dimerization to form HABI. In contrast, our results show that the preparation of lophine anions is not necessarily a prerequisite for dimerization as outlined in Fig. 4d. Mechanistically, the present electrochemical coupling reaction between lophines would involve two steps: (i) cathodic deprotonation into lophine anions due to the relatively acidic nature of imidazole protons and (ii) anodic oxidation into anion radicals and their spontaneous coupling into HABI (Fig. 4d).

**Electrochemical star to network topological transformation of polysiloxanes.** One ideal in synthetic chemistry is to develop a superconvenient reagentless recyclable chemical system for sustainable future. Taking into account that the present electrochemical oxidation reactions do not require inorganic chemical oxidants, a further sophisticated ETT system would be conceptually possible. Therefore, we utilized an ionic liquid as a support electrolyte doubled as the solvent, which we do not have to remove after the reaction because they have been utilized as an involatile environmentally benign solvent[27]. Moreover, we prepared a cap equipped with two mechanical pencil leads (Supplementary Fig. 8a) to enable a facile reaction even in a glass vial that can be found anywhere in the laboratories. Thus, three-armed star-shaped PDMS with lophine end groups ($S_L$) was synthesized (Supplementary Scheme 3a) and mixtures of solid state $S_L$ (Supplementary Fig. 8b) and 1-ethyl-3-methylimidazolium bis(trifluoromethanesulfonyl)imide ([EMI][TFSI]), silicone compatible ionic liquid[28], were prepared. Preliminary experiments showed that a slight liquid–liquid phase separation was observed in mixtures more than 30 wt% of [EMI][TFSI] and complete dissolution was difficult less than 10 wt%. Mixtures containing 10–30 wt% of [EMI][TFSI] resulted in a homogeneous oily liquid as represented by the fluidity of the mixture containing 30 wt% of [EMI][TFSI] (Supplementary Movie 1). Taking into account that the

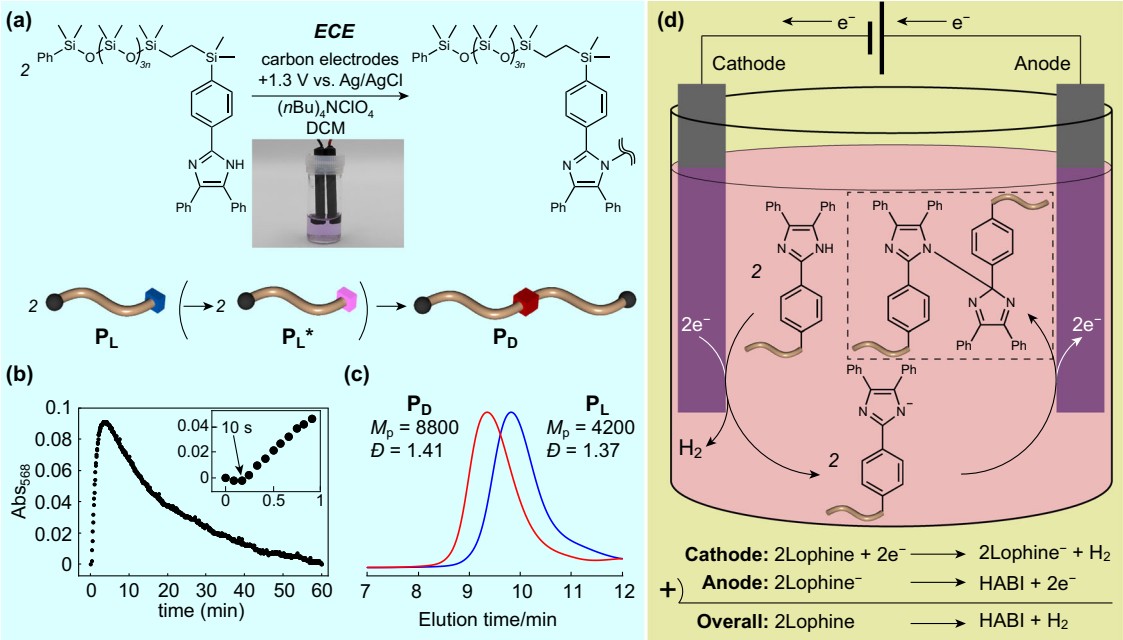

**Fig. 4 Electrochemical oxidation of $P_L$. a** Synthesis of $P_D$ upon applying a potential of 1.3 V to DCM solution of $P_L$ containing $(nBu)_4NClO_4$ with carbon electrodes. The inset photograph shows the pink-colored reaction solution upon applying potential. **b** Time-course plots of $A_{568}$ against time upon electrochemical oxidation of $P_L$. The inset is the magnification of the corresponding time range of 0–1 min. **c** SEC traces of $P_L$ (blue line) and $P_D$ (red line). **d** Schematic illustration of the mechanistic aspect with electrochemical oxidation of lophine-appended polymers.

electrochemical window of [EMI][TFSI] (the maximum applicable potential of ca. 1.0 V)[29], electrochemical oxidation of $S_L$ was performed. Upon applying a potential of 1.0 V, a colorless mixture of $S_L$ and [EMI][TFSI] (weight ratio: $S_L$/[EMI][TFSI] = 70/30) (Fig. 5a) readily turned pale gray with the formation of bubbles (Supplementary Movie 2), indicating the production of TPIRs and the release of hydrogen involved in the suggested reaction mechanism (Fig. 4d). Upon mild stirring with the electrodes, the mixture became highly viscous (Supplementary Movie 3) and finally became nonflowable (Supplementary Movie 4), even when the vial was inverted (Fig. 5b). As classical rubber theory tells us that network formation from liquid materials primarily reflects on their elasticity[30–32], the nonflowable behavior strongly suggests the production of network PDMS (**N**) swollen with [EMI][TFSI] (Supplementary Scheme 3b). To directly validate this, we next investigated the rheological properties of the resulting gel ($IN_{70}$) (weight ratio: **N**/[EMI][TFSI] =70/30). Frequency-dependent plots of storage and loss moduli ($G'$ and $G''$) measured for $IN_{70}$ indeed show higher $G'$ than $G''$ above 0.25 Hz (Fig. 5c), which is good indicative of solid materials ($G' > G''$)[31,33,34]. With the benefit of reversibility of HABI linkages in the network composing polymer chains, photocontrol of rheological properties would be appealing for ready-to-use materials. From apparent changes in the flowability of $IN_{70}$ upon photoirradiation (Supplementary Movie 5), we initially envisioned that both $G'$ and $G''$ would decrease upon photoirradiation. While $G''$ decreased as expected, $G'$ unexpectedly increased from ca. 130 kPa to 180 kPa upon one-time photoirradiation ($\lambda = 365$ nm, 150 mW/cm$^2$) as evident from the time-dependent analysis of $G'$ and $G''$ (Fig. 5d). In contrast, when we repeated ON–OFF cycles of photoirradiation with the same sample, $G''$ repeatedly decreased only during photoirradiation (Fig. 5e). When network formation takes place based on the dimerization of TPIRs generated by electrochemical oxidation of lophine end groups of $S_L$, nonuniform network topologies would be constructed due to solidification under a considerably small amount of the [EMI][TFSI] fraction. Given that more uniform network polymers generally exhibit better mechanical properties

than less uniform polymers[35,36], the reason for the increase in $G'$ upon the first one-time photoirradiation might be that the network topologies within the material became more uniform during the reversible cleavage of the covalent bonds in HABIs. The photoresponse of $G'$ observed in the following ON–OFF cycles of photoirradiation agreed with reported characteristics for cyclic PDMS linked with HABIs in the chains[4].

**Conclusions**. In summary, facile, efficient, and less wasted electrochemical coupling reactions and topological transformations of siloxane oligomers and polymers have been developed. The apparent change in the color of the reaction solution due to the production and consumption of TPIRs enables practically advantageous visual evaluation of the reactions. The ETT and ECE enabled by the present methodology are particularly important for controlling the properties of polymers. In the ETT system, a low effective concentration of reactive TPIR end groups is attained by the nature of the electrochemical reaction and enables predominant intramolecular cyclization. Moreover, the methodology enables the one step ETT of star-shaped PDMSs into their networks by utilizing an ionic liquid doubled as a green solvent and support electrolyte. The preclusion of chemical oxidants from the reaction system and replacement of volatile organic solvents with involatile environmentally benign ionic liquids are extremely important from the perspective of sustainability. It is also worth mentioning that the present methodology enables ETT-based network polymer synthesis only by applying a voltage of 1 V to mechanical pencil leads. In other words, anyone with pencils and a battery can perform ETT with our materials. Encouraged by the global demand for pursuing sustainability, the present methodology will open a new avenue for the development of innovative materials fabricated with environmentally-benign electrochemical processes.

## Methods

**Electrochemical oxidation of 1.** In a typical procedure, the known compound $2$[21] was synthesized by electrochemical oxidation of **1**. Thus, DCM/MeOH (5/2, v/v)

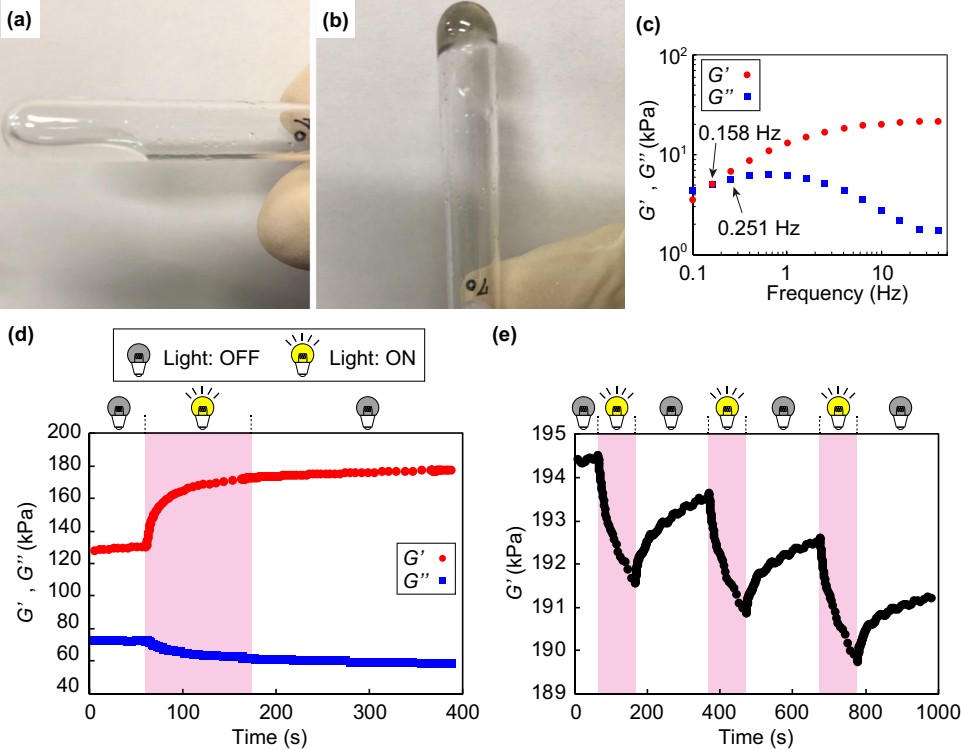

**Fig. 5 Star-to-network electrochemical topological transformation. a, b** Photographs of a mixture of **S_L** and [EMI][TFSI] (weight ratio: **S_L**/[EMI][TFSI] =70/30) (**a**) and **IN_70** (**b**). **c** Frequency-dependent plots of $G'$ and $G''$ of **IN_70** under the strain of 1%. **d** Time-course plots of $G'$ and $G''$ upon one-time photoirradiation (365 nm, 150 mW/cm²) to **IN_70**. **e** Time-dependent plots of $G'$ upon ON–OFF cycles of photoirradiation to a sample after one-time photoirradiation to **IN_70**. Photoirradiation was performed during time ranges indicated with pink shading and the measurements were conducted under the strain of 1% at the frequency of 5 Hz.

solution (7 mL) of **1** (375 mg, 1.0 mmol) and (nBu)₄NClO₄ (342 mg, 1.0 mmol) was prepared into an IKA ElectraSyn vial, and the current was passed at 1.25 V constant potential with carbon electrodes. After 2 h, aliquots taken from the mixture were subjected to mass spectroscopic analyses. APCI-MS calculated for $C_{42}H_{28}Br_2N_4$ [M + H]⁺ 747.1, found 747.5. For ¹H NMR analysis, the mixture was concentrated and the residue was dissolved in MeOH and precipitated in H₂O to afford the product in 72% yield. ¹H NMR (500 MHz, DMSO-$d_6$) δ ppm 7.10–8.30 (Ar*H*).

**Electrochemical oxidation of M_L and P_L**. Into an IKA ElectraSyn vial was prepared DCM solution (5 mL) of **M_L** (20 mg) and (nBu)₄NClO₄ (136 mg), and the current was passed at 1.25 V constant potential with carbon electrodes for 3.5 h. After the removal of the volatile fraction by evaporation, the residue was diluted with hexane, and filtered, and the filtrate was concentrated to dryness to afford **M_C** in quantitative yield. The ¹H NMR spectrum (Supplementary Fig. 5b) matched with the reported one[23]. The $M_n$, $M_w$, $M_p$, and Đ measured by SEC calibrated with polystyrene standards were $M_n = 1400$, $M_w = 1600$, $M_p = 1300$, and Đ =1.16, respectively. HR-MS (ESI+) calculated for $C_{60}H_{78}N_4O_6Si_7$ [M + H]⁺ 1147.4384, found 1147.4458. Likewise, electrochemical oxidation of **P_L** was performed and characterized based on ¹H NMR spectrometry (Supplementary Fig. 7b). The $M_n$, $M_w$, $M_p$, and Đ of the product **P_D** measured by SEC calibrated with polystyrene standards were $M_n = 6200$, $M_w = 8700$, $M_p = 8800$, and Đ =1.41, respectively.

**Electrochemical oxidation of S_L**. Into a vial, **S_L** (70 mg) and [EMI][TFSI] (30 mg) were added, and the vial was closed with a plastic cap equipped with mechanical pencil leads as carbon electrodes. Current was passed at 1.00 V constant potential until gelation.

**Cyclic voltammetry measurements**. Cyclic voltammograms were recorded on an IKA ElectraSyn 2.0 Pro in DCM/MeOH (10/3, v/v) containing 0.01 M analyte and 0.1 M (nBu₄)NClO₄ by using glassy carbon as working electrodes, Pt as a counter electrode, and Ag/AgCl in 3 M KCl in H₂O as a reference electrode.

**Rheological analyses**. Rheological analyses were performed on an Anton Paar MCR 102 rheometer equipped with a Pertier temperature control device and a photocuring system using a parallel plate with a diameter of 12 mm on a glass plate. Frequency-dependent analyses of $G'$ and $G''$ were performed at 25 °C with the

sample thickness of 0.20 mm. Time-dependent analyses of $G'$ and $G''$ upon one-time photoirradiation (365 nm, 150 mW/cm²) was performed with the sample thickness of 0.30 mm and those upon ON–OFF cycles of photoirradiation (365 nm, 150 mW/cm²) were perfomed with the sample thickness of 0.20 mm at the frequency of 5 Hz at 25 °C. Photoirradiation was performed on a Hamamatsu LIGHTNING CURE LC8 L9588 with a Hamamatsu A10014-35-0110 light guide. A Hamamatsu A9616-07 filter was used for transmitting light with the wavelength at 365 nm.

## Data availability

All data supporting the findings of this study are available within the article (and Supplementary Information File(s)) or available from the corresponding author on reasonable request.

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

## Acknowledgements

This work was supported by JSPS KAKENHI (Grant Numbers 20K21218 and 21H01632, S.H.), The University of Tokyo Excellent Young Researcher Program (S.H.), and the TEPCO Memorial Foundation, Research Grant (Basic Science) (S.H.). We are grateful to our industrial collaborators for financial support. We also thank Prof. Taro Toyota (The University of Tokyo) for his kind support.

## Author contributions

M.O. and S.H. conceived the project and designed the experiments. M.O. performed synthesis and characterization. S.H. and M.O. wrote the manuscript. S.H. directed the project.

## Competing interests

The authors declare no competing interests. Satoshi Honda is an Editorial Board Member for *Communications Chemistry*, but was not involved in the editorial review of, or the decision to publish this article.
