## [Peer Review File · Communications Chemistry]

Reviewers' comments:

Reviewer #1 (Remarks to the Author):

In their manuscript Oka and Honda report the unprecedented electrochemical dimerization of TPI radicals into HABI in the context of macromolecular transformations. This is a synthetically unique and important approach to HABI synthesis since established oxidation methods using potassium ferricyanide produce liters over liters of cyanide-containing solvent waste. From experience this is, by the way, not a mild annoyance; it is a serious logistic and safety concern when working on HABI synthesis on material scale-relevant batches. The authors then show how they use their developed procedure to perform topological transformations and chain extensions, which also wasn't carried out before. Another novel and noteworthy aspect is the use of ionic liquids as generally the oxidation conditions are not compatible with solvents the polymers are soluble in. In general, this work is important from the standpoint that it offers a new method to generate responsive macromolecular architectures that bear the HABI motif.

In II.67 the authors indicate that they used EPR and APCI-MS to prove the HABI formation on the small molecules. However, no quantitative information can be gathered from this. Did the authors run proton NMR to conclusively support their claim? I know this is hard to analyze as the dimerization is not as straightforward as depicted in Fig. 2a and actually an isomer mixture is formed, but it can be done. On this note, all measured NMR (also the macromonomers after hydrosilylation, e.g. ML and MC!) and mass spectra should be shown as supporting figures in the supplementary information. On the macromolecular topological transformations, the GPC data are very much indicative of a near-quantitative oxidation and dimerization to HABI but only NMR can be considered as final proof.

Similarly, the authors describe a home-built setup for spectroelectrochemistry in II. 123. For the measurements in Fig. 4b, the recorded spectra over time should be shown in the SI.

In the context of polymer chemistry, the authors might find the works of Scott and coworkers (e.g. <https://doi.org/10.1016/j.dental.2015.06.005>) interesting to further strengthen their point about the usefulness of their system. Moreover, there is a whole catalogue of papers about HABI and HABI-like photoswitches from the group of Jiro Abe. The authors might consider revising their introduction to clearly highlight the importance of HABI as responsive molecules. Only using references 2 and 4 appears a little "underselling" in this regard.

Concludingly, I find the work highly interesting, logically structured, and relevant but only recommend publication after the complete characterization data has been included in the SI.

Minor remarks:

II. 180: Fig. 4c should likely be Fig. 5c.

Fig. 5c-e: The caption should state the employed strain amplitudes and frequencies where applicable.

Fig 5e: The caption should likely state that the data is after the "virgin cycle" from Fig. 5d, otherwise the panels d and e seem to directly contradict each other.

Reviewer #2 (Remarks to the Author):

- 1) This paper is not first paper or original paper for electrochemically topology-controlled polymerization.
- 2) This paper does not solve general question how to achieve high molecular weight by electrosynthesis.
- 3) In introduction of this paper "Although various polymer–polymer coupling reactions have been utilized for synthesizing nonlinear polymers, development of an efficient and greener approach still remains challenging." There is no any reference, which is should be very important to evaluate the appealing of this paper. I think this paper is not well-prepared to be submitted.

Reviewer #3 (Remarks to the Author):

Oka et al. describes an electrochemical method to synthesize polymeric structures with different topologies strating from 2,4,5-triphenylimidazole (lophine)-terminating polymers and oligomers. This process enables the preparation of low-molecular-weight cyclic polymers, multimers and polymer networks (starting from stars polymers).

This topic is of high interest, and the reviewer believes it is of interest for this journal. The results and characterizations presented by the authors are convincing and rather accurate. The paper is well written, although the aauthors should double-check the use of certain words (such as "pestilent" on page 6 line 84, maybe "deleterious" would be more appropriate).

Among the references, the authors should include those works where alternative and efficient topological transformations were described. Among several works one could be Sumerlin et al. *Nat. Chem.* 2017, 9(8), 817-823. Overall, this work deserves publication in *Commun. Chem.* after minor revisions.

Response to Referees

First of all, we thank the editor and reviewers very much for the valuable comments and suggestions. We have thoroughly reviewed all of the editor's and reviewers' comments and have elaborated the manuscript.

Reviewer #1:

In their manuscript Oka and Honda report the unprecedented electrochemical dimerization of TPI radicals into HABI in the context of macromolecular transformations. This is a synthetically unique and important approach to HABI synthesis since established oxidation methods using potassium ferricyanide produce liters over liters of cyanide-containing solvent waste. From experience this is, by the way, not a mild annoyance; it is a serious logistic and safety concern when working on HABI synthesis on material scale-relevant batches. The authors then show how they use their developed procedure to perform topological transformations and chain extensions, which also wasn't carried out before. Another novel and noteworthy aspect is the use of ionic liquids as generally the oxidation conditions are not compatible with solvents the polymers are soluble in. In general, this work is important from the standpoint that it offers a new method to generate responsive macromolecular architectures that bear the HABI motif.

Response

We are grateful to the reviewer for detailed and careful evaluation on our manuscript. We have thoroughly reviewed the comments and reexamined whole of our manuscript and presented important data including ^1H NMR.

Comment #1

In ll.67 the authors indicate that they used EPR and APCI-MS to prove the HABI formation on the small molecules. However, no quantitative information can be gathered from this. Did the authors run proton NMR to conclusively support their claim? I know this is hard to analyze as the dimerization is not as straightforward as depicted in Fig. 2a and actually an isomer mixture is formed, but it can be done. On this note, all measured NMR (also the macromonomers after hydrosilylation, e.g. ML and MC!) and mass spectra should be shown as supporting figures in the supplementary information. On the macromolecular topological transformations, the GPC data are very much indicative of a near-quantitative oxidation and dimerization to HABI but only NMR can be considered as final proof.

Response

We have presented ^1H NMR spectra of small molecular HABI (product: **2**), **M_L**, **M_C**, **P_L**, and **P_D** according to the suggestion. As was pointed, dimerization afforded several isomers of HABIs but disappearance of signals derived from lophine precursors has been clearly confirmed based on ^1H NMR. We are grateful to this comment and we believe that our manuscript has been significantly improved based on the comment.

Comment #2

Similarly, the authors describe a home-built setup for spectroelectrochemistry in II. 123. For the measurements in Fig. 4b, the recorded spectra over time should be shown in the SI.

Response

We have described the home-built setup for spectroelectrochemistry in the supporting information and the relevant supplementary figures have also been updated. Regarding Fig. 4, we have measured absorbance only at the wavelength of 568 nm and no recorded spectra over time are available. Since TPIR reacts quickly with another TPIR, we found that if we measure the spectrum in a sufficient wavelength range for detecting TPIR absorption, the maximum TPIR absorption significantly changes during the time required for the measurement, making it unreproducible. Therefore, we performed adsorption measurements only at the wavelength of 568 nm in Fig. 4b.

Comment #3

In the context of polymer chemistry, the authors might find the works of Scott and coworkers (e.g. <https://doi.org/10.1016/j.dental.2015.06.005>) interesting to further strengthen their point about the usefulness of their system. Moreover, there is a whole catalogue of papers about HABI and HABI-like photoswitches from the group of Jiro Abe. The authors might consider revising their introduction to clearly highlight the importance of HABI as responsive molecules. Only using references 2 and 4 appears a little “underselling” in this regard.

Response

We are grateful to this comment. We have cited the publications from the group of Scott and Abe and their coworkers. Also, we have discussed more about related references in the Introduction of the revised manuscript.

Comment #4

Concludingly, I find the work highly interesting, logically structured, and relevant but

only recommend publication after the complete characterization data has been included in the SI.

Response

We believe that we have presented thorough characterization of our materials in the updated version of our manuscript and supplementary information.

Comment #5

Minor remarks:

ll. 180: Fig. 4c should likely be Fig. 5c.

Fig. 5c-e: The caption should state the employed strain amplitudes and frequencies where applicable.

Fig 5e: The caption should likely state that the data is after the “virgin cycle” from Fig. 5d, otherwise the panels d and e seem to directly contradict each other.

Response

We have updated these points in the revised version of our manuscript.

Reviewer #2:

Comment #1

1) This paper is not first paper or original paper for electrochemically topology-controlled polymerization.

Response

Our target in the present study is not polymerization reactions but polymer-polymer coupling reactions based on electrochemistry. We believe that all of present electrochemical topological transformation, chain extension, and network formation reactions described in the manuscript are first and original.

Comment #2

2) This paper does not solve general question how to achieve high molecular weight by electrosynthesis.

Response

If the target of a research is the electropolymerization, then synthesis of high-molecular-weight polymers would be important to test generality or limitation of such polymerization reactions. However, as with the response to the previous comment, the target in the present study is not polymerizations but polymer-polymer coupling reactions and we have achieved our targeted electrochemical coupling reactions as

described in the manuscript.

Comment #3

3) In introduction of this paper “Although various polymer–polymer coupling reactions have been utilized for synthesizing nonlinear polymers, development of an efficient and greener approach still remains challenging.” There is no any reference, which is should be very important to evaluate the appealing of this paper. I think this paper is not well-prepared to be submitted.

Response

As the importance of polymer–polymer coupling reactions in synthesizing nonlinear polymers or topological transformation has been described in several reviews, we have updated references relating to topological transformation (Refs. 19 and 20) in the updated version of our manuscript.

Reviewer #3:

Oka et al. describes an electrochemical method to synthesize polymeric structures with different topologies starting from 2,4,5-triphenylimidazole (lophine)-terminating polymers and oligomers. This process enables the preparation of low-molecular-weight cyclic polymers, multimers and polymer networks (starting from stars polymers).

This topic is of high interest, and the reviewer believes it is of interest for this journal. The results and characterizations presented by the authors are convincing and rather accurate. The paper is well written, although the authors should double-check the use of certain words (such as "pestilent" on page 6 line 84, maybe "deleterious" would be more appropriate).

Response

We have strongly encouraged by the comments. We have thoroughly reviewed wording, grammar, and “a” or “the” and have updated our manuscript.

Among the references, the authors should include those works where alternative and efficient topological transformations were described. Among several works one could be Sumerlin et al. Nat. Chem. 2017, 9(8), 817-823. Overall, this work deserves publication in Commun. Chem. after minor revisions.

Response

According to the suggestion, references including Nat. Chem. 2017, 9(8), 817-823 have been cited and the related discussion has been updated in the revised version of our manuscript.

REVIEWERS' COMMENTS:

Reviewer #1 (Remarks to the Author):

With their revised manuscript Oka and Honda solve all questions and remarks raised by me to a satisfactory degree. The authors have added convincing characterization data in the form of the requested $^1\text{H-NMR}$ spectra and experimental details for spectroelectrochemistry. It is understandable that the authors cannot provide a full time series of UV-vis spectra due to the fast reaction and hence chose to measure in kinetics mode at a single wavelength. The revision of the introduction now gives the manuscript a wider scope. Concludingly, I can recommend the publication in its current form.

Reviewer #2 (Remarks to the Author):

I still doubt the originality and significancy, but I would like to recommend it for publication.

Response to Referees

Reviewer #1:

With their revised manuscript Oka and Honda solve all questions and remarks raised by me to a satisfactory degree. The authors have added convincing characterization data in the form of the requested $^1\text{H-NMR}$ spectra and experimental details for spectroelectrochemistry. It is understandable that the authors cannot provide a full time series of UV-vis spectra due to the fast reaction and hence chose to measure in kinetics mode at a single wavelength. The revision of the introduction now gives the manuscript a wider scope. Concludingly, I can recommend the publication in its current form.

Response

We are grateful to the Reviewer #1. We were strongly encouraged by the comments from Reviewer #1 and we believe that our manuscript has greatly been improved based on the comments and suggestions. We appreciate the Reviewer #1 very much for recommending our manuscript for publication.

Reviewer #2:

I still doubt the originality and significancy, but I would like to recommand it for publication.

Response

We thank the Reviewer #2 for reviewing our revised manuscript. We are open to any feedback from the editor and reviewers and are willing to work on improving our manuscript. We believe that the revised version of our manuscript fully expresses the originality and significance of our work. We are delighted that you finally recommended our work for publication.